# Applicability of Obesity-Related SNPs and Their Effect Size Measures Defined on Populations with European Ancestry for Genetic Risk Estimation among Roma

**DOI:** 10.3390/genes11050516

**Published:** 2020-05-06

**Authors:** Erand Llanaj, Péter Pikó, Károly Nagy, Gábor Rácz, Sándor János, Zsigmond Kósa, Szilvia Fiatal, Róza Ádány

**Affiliations:** 1Public Health Research Institute, University of Debrecen, Kassai St 26/B, H-4028 Debrecen, Hungary; erand.llanaj@sph.unideb.hu; 2Doctoral School of Health Sciences, University of Debrecen, Kassai St 26/B, H-4028 Debrecen, Hungary; 3MTA-DE-Public Health Research Group, Public Health Research Institute, University of Debrecen, Kassai St 26/B, H-4028 Debrecen, Hungary; piko.peter@sph.unideb.hu; 4Department of Preventive Medicine, Faculty of Public Health, University of Debrecen, Kassai St 26/B, H-4028 Debrecen, Hungary; nagy.karoly@sph.unideb.hu (K.N.); racz.gabor@sph.unideb.hu (G.R.); sandor.janos@sph.unideb.hu (S.J.); fiatal.szilvia@sph.unideb.hu (S.F.); 5Department of Methodology for Health Visitors and Public Health, Faculty of Health, University of Debrecen, Sóstói street 2-4, 4400 Nyíregyháza, Hungary; kosa.zsigmond@foh.unideb.hu

**Keywords:** obesity, single nucleotide polymorphisms, genetic risk score, susceptibility, Roma, Hungary

## Abstract

Investigations on the impact of genetic factors on the development of obesity have been limited regarding the Roma population—the largest and most vulnerable ethnic minority in Europe of Asian origin. Genetic variants identified from genetic association studies are primarily from European populations. With that in mind, we investigated the applicability of data on selected obesity-related single nucleotide polymorphisms (SNPs), obtained from the Hungarian general (HG) population of European origin, on the Hungarian Roma (HR) population. Twenty preselected SNPs in susceptible alleles, known to be significantly associated with obesity-related phenotypes, were used to estimate the effect of these SNPs on body mass index (BMI) and waist circumference (WC) in HG (*N* = 1783) and HR (*N* = 1225) populations. Single SNP associations were tested using linear and logistic regression models, adjusted for known covariates. Out of 20 SNPs, four located in *FTO* (rs1121980, rs1558902, rs9939609, and rs9941349) showed strong association with BMI and WC as continuous variables in both samples. Computations based on Adult Treatment Panel III (ATPIII) and the International Diabetes Federation’s (IDF) European and Asian criteria showed rs9941349 in *FTO* to be associated only with WC among both populations, and two SNPs (rs2867125, rs6548238) in *TMEM18* associated with WC only in HG population. A substantial difference (both in direction and effect size) was observed only in the case of rs1801282 in *PPARγ* on WC as a continuous outcome. Findings suggest that genetic risk scores based on counting SNPs with relatively high effect sizes, defined based on populations with European ancestry, can sufficiently allow estimation of genetic susceptibility for Roma. Further studies are needed to clarify the role of SNP(s) with protective effect(s).

## 1. Introduction

Roma constitute the largest ethnic minority population in Europe (estimated to be between 12 and 16 million) [1] and have been a major focus of ethnicity-based studies in past decades [2]. The current hypothesis argues that Roma, living mainly in Europe, migrated from South Asia through the Balkans to European countries more than a thousand years ago [3,4]. Recent genetic studies investigating the origins of Roma have suggested the Northwest region of India as the source of their South Asian ancestry [5,6,7,8]. However, the ancestral group or geographic region within South Asia that is most closely related to the ancestral population of the Roma is still a subject of speculation [9]. 

Generally, health and well-being statistics of the Roma minority appear relatively worse, compared to that of the host majority populations, independently of the country where they live, usually in disadvantaged conditions (frequently in segregated colonies) [10,11,12,13,14]. Hungary is one of the countries in which the representation of Roma is over 8% of the total population, and this proportion is likely to increase in the near future due to high population growth rate among Roma and the opposite, i.e., a decreasing fertility rate, among the majority population [15]. 

Recent research involving samples from the adult Hungarian general (HG) population, as well as from Hungarian Roma (HR) communities, has shown a high prevalence of metabolic syndrome in the HG population (34.95%, 95% confidence interval (CI) 32.57–37.33) and an even higher prevalence among HR (36.38%, 95% CI 32.67–40.09) [16]. However, no notable differences in the prevalence of central obesity has been observed between the two groups. 

Our most recent survey revealed that the prevalence of obesity has drastically and significantly increased among younger Roma (for both sexes) over the last decade [2]. Obesity was shown to disproportionally affect Roma, as the frequency of obesity among HR females aged 18–29 years old has become almost four times compared to that of the same age group in the HG population. In addition, a recent report ranked Hungary as the fourth most obese country among Organization for Economic Co-operation and Development (OECD) countries, with more than 30% of adults being obese in 2017 [17]. 

Obesity is an important common precondition of major noncommunicable diseases with a multifactorial aetiology [18], i.e., in addition to biological, behavioural, and socio-economic conditions, genetic factors are also important contributors [19,20]. Recently, obesity rates have escalated to a global pandemic with varying prevalence across ethnic groups [21]. These differences are partially explained by lifestyle factors, especially unhealthy diet and insufficient physical activity, in addition to the genetic predisposition for obesity. Elucidating genetic aspects, as well as the metabolic pathways they influence, can significantly help in addressing obesity within a more precise public health approach [22], and applying such an approach necessitates a population-based, integrated system of genomic risk estimation and/or prediction [22,23,24]. In this context, the number of genomic investigations estimating the role of genetic factors in the increased susceptibility to obesity is abundant in European-ancestry populations [25], but the number of studies among Roma is extremely limited and restricted to the detection of associations between obesity phenotypes and certain selected single nucleotide polymorphisms (SNPs) on extended Roma families [26,27]. 

Previously, the cumulative effect of 20 SNPs has been estimated using unweighted and weighted genetic risk score (GRS and wGRS) models, with the main aim to compare the genetic load of HR and HG populations for obesity [28]. Overall, results showed no significant differences in terms of the genetic load and this may indicate that the difference in obesity prevalence between these two populations may not be explained solely by the differences in their genetic susceptibility. However, we hypothesized that these estimates of effect sizes (so called weights) used for wGRS computation may not be applicable for risk estimation in Roma, considered to have an Asian origin, as they were defined based on European populations. Therefore, a question arises on the transferability of effect size estimates obtained from European populations, i.e., whether such data can be utilized to assess the genetic risk of Roma. 

To answer this question, individual effects of a set of twenty preselected SNPs on Hungarian general (with European origin) and on Roma (with supposedly Asian origin) samples were estimated and compared in case of different obesity and abdominal obesity classifications and phenotyping criteria.

## 2. Material and Methods

### 2.1. Study Design

The study included samples and data assembled during recent cross-sectional surveys including 1260 Roma individuals living in segregated colonies in North-East Hungary (where this minority population is mainly concentrated), and 1819 individuals representing the Hungarian general population [16,29]. The source population for the HG subjects included all individuals aged 20–69 years who were registered by general practitioners (GPs) involved in the General Practitioners’ Morbidity Sentinel Stations Program [30] of eight Hungarian counties. All participants were invited to take part in the study by their GPs, who performed physical examinations (weight, height, blood pressure measurements, etc.) and collected blood samples for laboratory investigations and DNA isolation. 

### 2.2. Sampling of Hungarian Roma Subjects

A nationwide project surveyed the segregated colonies (SCs) in Hungary [31]. Roma field workers nominated by Roma non-governmental organizations identified the SCs. In all, 94% of the SCs’ inhabitants declared themselves to be HR. The colony registry was used for the stratified multistep sampling. Both the first and second Roma survey subjects were sampled from two Hungarian counties (Hajdu-Bihar and Szabolcs-Szatmar-Bereg), where Roma colonies are largely concentrated. Since the focus of our investigation was the highly segregated, closed Roma population, SCs with more than 100 inhabitants were considered as the study base. Out of the 64 eligible SCs, 40 and 25 households from each SC were randomly selected using the GPs’ validated household lists. Only individuals between 20–69 years old from the resulting 1000 households comprised the final sampling frame, and one person from each household was chosen by a member of the primary health-care team using a random table. In this way, the final samples were representative of the general HR population living in the designated areas of interest. Following this project, there were two surveys, one in 2011 [16] and the other in 2013 [29], which collected data on HR subjects (Appendix A).

#### 2.2.1. Recruitment of Hungarian Roma Adult Subjects 

HR participants, in both surveys, were invited into the GPs’ offices, where GPs or practice nurses would complete a questionnaire on socio-demographic factors, lifestyle, and self-assessed health status on the basis of interviewees’ answers, and physical examination was carried out. Subjects who did not sign the written consent were excluded from both surveys. The health status description used the former medical records of participants as well. Blood samples for laboratory investigations were taken. The data collection of the first survey started in September 2011 and took 4 months, and it applied the methods of a former study on metabolic syndrome [31]. The second data collection was conducted in August 2013 and took 1 month.

#### 2.2.2. Recruitment of Hungarian General Adult Subjects

HG subjects were sampled from the first survey [16], and the source population included all individuals aged 20–69 years who were registered by the 59 participating GPs of eight Hungarian counties in the framework of the General Practitioners’ Morbidity Sentinel Stations Program [31]. The study population was representative of the larger HG population as it was randomly selected proportional to the size of the practices in order to represent the Hungarian adult population with regards to geographic, age, and sex distributions. More details are given in Appendix A, and all relevant demographic data for HR subjects from both surveys and for HG subjects included in this analysis are given in Appendix A.

### 2.3. Genotyping and Anthropometric Data

In the present study, we examined and estimated obesity risk for the twenty SNPs investigated previously [28] (Appendix A). Frequency distributions of risk alleles with SNPs specified in the two study populations were compared using binomial distribution and Kolmogorov–Smirnov tests, respectively. General obesity risk was determined by comparing genotypes (namely the frequency of risk alleles) of subjects with body mass index (BMI) less than 25 kg/m^2^ (as a reference group) with that of subjects with BMI ≥ 30 kg/m^2^ in both populations. 

Abdominal obesity was determined using waist circumference (WC) standards, defined for females and males by the Adult Treatment Panel III (ATPIII) criteria [32] and the International Diabetes Federation (IDF), separately for European (IDF_EURO_) and Asian (IDF_ASIA_) populations [33]. Because of the Asian origin of Roma and the European origin of the HG population, results obtained based on IDF_ASIA_ criteria for the Roma and IDF_EURO_ for HG were considered as relevant.

Appraisal of genotypes’ effect on the quantitative traits (BMI and WC) involved linear and logistic regression analyses (under additive model) were performed to obtain effects of genotypes on dichotomous traits (generalized and abdominal obesity). SNPs with significant effect size measures (i.e., *β* and odds ratio (OR)) were used as a model for investigating the extent of transferability of SNPs’ effect size measures of obesity phenotypes from HG to HR. Five SNPs with significant effect were examined for comparisons on obesity and WC as continuous outcomes, and nine SNPs for comparison on obesity and WC as binary outcomes. Complete results for all twenty SNPs can be found in the Appendix A). 

Additionally, SNPs with significant effect measures for BMI and WC as a continuous outcome were charted and color-coded to visualize the findings on the differences and overlaps between the two study populations under investigation (Appendix A). Genotyping and quality control were performed by a service provider (i.e., Mutation Analysis Core Facility of the Karolinska University Hospital, Stockholm, Sweden). Additional details on the study design, sampling, DNA isolation, SNPs selection process, and genotyping are described in detail elsewhere [28]. 

### 2.4. Ethical Statement 

The study was approved by the Ethical Committee of the University of Debrecen, Medical Health Sciences Centre (reference No. 2462–2006) and by the Ethical Committee of the Hungarian Scientific Council on Health (reference Nos. NKFP/1/0003/2005; 8907-O/2011-EKU). All procedures were performed in accordance with the ethical standards of the institutional and national research committee, as well as with the 1964 Helsinki declaration and its later amendments or comparable ethical standards. All subjects gave their written informed consent for the study and no minors were involved. 

### 2.5. Statistical Analyses

Samples with incomplete genotype and phenotype data (i.e., missing BMI and WC) were excluded from the current analysis (the number of excluded records in the HR and HG populations was 119 and 247, respectively). Transformations and the rest of statistical analyses were carried out using Stata version 13.1 (StataCorp LLC, Lakeway, TX, USA) and SPSS version 21 (IBM, Armonk, NY, USA). The Shapiro–Wilk test was performed to check assumptions of normality. 

Dependent quantitative variables such as BMI, WC, and age were transformed using a two-step approach suggested by Templeton in order to reduce effects of non-normality [34]. Student’s *t*-tests were used to compare age, BMI, and WC differences, while the chi-square (χ^2^) test was used for comparing sex distribution between populations. Allele frequencies of the study populations were calculated and compared. The Hardy–Weinberg equilibrium (HWE) was evaluated by the χ^2^ test and no significant deviation from HWE was observed in the study populations. Linkage disequilibrium (LD) was estimated by Haploview software (version 4.2) [35]. 

The association of SNPs with different outcomes were measured individually. This calculated effect was independent from the inserted covariates, but not independent from those SNPs’ effect, which shared the same LD block (Appendix A). In the case of further examination of the combined effect of these SNPs, the degree of LD between them should be considered. 

In addition, regression analyses were used to further test the association of SNPs with obesity. Association analyses were also performed assuming co-dominant, dominant, recessive, and over-dominant genetic models. Akaike information criterion (AIC) and Bayesian information criterion (BIC) were used for model selection. In our analysis, the most frequent homozygote genotype was considered as the reference (Appendix A–S7). 

In order to create a better and more accurate understanding and estimation toward the population-specific impact of the most notable and differing variant (i.e., rs1801282 (*PPARγ*)) toward susceptibility to obesity of HR, we performed a meta-analysis of South Asian and Indian populations, which is the alleged origin of the HR population (see Section ‘Meta-analysis for *PPARγ* (rs1801282)’, in the Appendix A).

All results were adjusted for relevant covariates (i.e., age, sex). Power calculations were performed in both study groups by the software package Quanto version 1.2.4 [36] using the SNP-specific effective sample size of this study. The effect of SNPs was individually calculated with a nominal significance level of 5% (*p* = 0.05), and multiple comparison correction was applied when the SNPs’ effect was compared between the two populations (*p* = 0.0025).

## 3. Results

Out of the twenty SNPs involved in the present analysis, nine showed no significant association with any obesity- or WC-related phenotype in neither of the populations under investigation (namely rs10938397 (*GNPDA2*), rs1137101 (*LEPR*), rs1501299, rs2241766 (*ADIPOQ*), rs17782313 (*MC4R*), rs6265, rs925946 (*BDNF*), rs659366, and rs660339 (*UCP2*)). 

### 3.1. Characteristics of the Study Samples

As shown in Table 1, a total of 2637 subjects were included in the analysis, 1496 and 1141 from the HG and HR populations, respectively. There was a higher proportion of females in the HR population compared to that of the HG population. The mean age of the HG group was higher than that of HR group, thus all calculations were adjusted for sex and age. Mean values for BMI and WC and the prevalence of central obesity (according to ATPIII and European and Asian criteria of IDF) were higher in the case of the HG population.

In addition, the HG population displayed a consistently higher frequency of central obesity, as well as larger waist size and BMI compared to the HR sample, independently of classification applied for abdominal/central obesity.

### 3.2. Individual Effects of SNPs in the Hungarian General and Roma Population

The effect of the SNPs exhibiting significant associations with BMI and/or WC (as continuous outcomes) separately for each study population are shown in Table 2. Altogether, four SNPs (rs1121980, rs1558902, rs9939609, and rs9941349) located in the fat mass and obesity-associated (*FTO*) gene showed significant association with WC (as a continuous outcome) in both populations. The same four SNPs also showed a significant positive association with BMI in the HG population, and only two of them (rs9939609 and rs9941349) in the Roma population. 

The rs1801282, located in the peroxisome proliferator-activated receptor gamma (*PPARγ*) gene, was significantly associated with decreased waist circumference as a continuous outcome only in the HR population (HG: *β* = 0.340, SE: 0.724, *p* = 0.638; HR *β* = –3.507, SE: 1.699, *p* = 0.039; *p* = 0.029 for difference in association). This SNP had a large protective effect against increased WC but did not have any significant effect on BMI in the HR population. A protective effect of this extent was not observed for any of the obesity phenotypes in the HG sample. Effect sizes for WC were consistently stronger in both populations, compared to generalized obesity, suggesting that the risk for obesity in both populations is primarily dependent on abdominal obesity.

Further comparisons (Figure 1) were made by computing ORs for obesity (Figure 1A), WC by ATPIII (Figure 1B) and WC by IDF_EURO_ (Figure 1C), and IDF_ASIA_ (Figure 1D). When comparing associations of SNPs for obesity (Figure 1A), all four *FTO* variants displayed a similar significant association in both the HG and HR groups. rs16139 in the neuropeptide Y gene (*NPY*) had a significant effect on obesity (Figure 1A) only in the HG, while all other SNPs showed no association with obesity in either population. In the case of the WC ATPIII phenotype (Figure 1B), three SNPs (rs1558902, rs9939609, and rs9941349 in *FTO*) showed a significant effect in HR population (Figure 1B) and one (rs12970134 which is localized near the melanocortin 4 receptor (*MC4R*) gene), had a significant effect on WC in the HG population (Figure 1B).

Comparisons based on IDF_EURO_ (Figure 1C) showed two SNPs (rs2867125 and rs6548238—located in the transmembrane protein 18 (*TMEM18*) gene) significantly associated with WC in HG. In the case of IDF_ASIA_ (Figure 1D), no significant association was observed in HR. A summary of SNPs associated with obesity- and WC-related phenotypes as continuous outcomes, is presented in the supplementary Appendix A.

### 3.3. The Results of Different Genetic Models for the rs1801282 Gene Polymorphism in PPARγ

We used four genetic models (i.e., codominant, dominant, recessive, and over-dominant) to further examine the effect of the rs1801282 in *PPARγ* on the WC as a continuous outcome (Table 3). There was a significant association between SNP and outcome in the Roma population with a dominant genetic model. Furthermore, it was found that the direction of the effect of the rs1801282 differed in both study populations regardless of the genetic model. No significant difference in the risk susceptibility for this SNP was observed, with or without obesity phenotypes among HG or HR, when considering other generalized and centralized obesity criteria in further genetic models (Appendix A).

For *PPAR*γ SNP (rs1801282) the allelic frequency was significantly higher among Roma (OR = 2.74, 95% CI: 2.18–3.46), with ethnicity status as an outcome. 

## 4. Discussion

There is currently limited information on the relevance of the selected risk loci for estimating genetic risk for obesity among Roma, as most of the research has been primarily focused on populations of European ancestry. The role of different genetic factors in determining susceptibility to obesity for a defined population cannot be accurately estimated by using data obtained in one population group and then deploying these data in another sample, particularly when ancestral profiles are assumed to differ [37]. Considering the current hypothesis that the Roma population originally descends from South Asia, it is reasonable to presume that the available data in the literature need to be cautiously considered for genetic risk computations. Our findings suggest that effect size measures of the most important obesity-related SNPs, even if they were defined based on European populations, are applicable in studies designed to estimate the obesity risk among Roma. 

Results obtained in the present study revealed four SNPs (rs1121980, rs1558902, rs9939609, rs9941349) in *FTO* (which are in LD, see Appendix A) with a significant effect on obesity among Roma; two of which (namely rs9939609 and rs9941349 in *FTO*) had a significant effect on BMI and five (rs1121980, rs1558902, rs9939609, and rs9941349 in *FTO*, and rs1801282 in *PPARγ*) on WC. All of these SNPs belong to the same highly correlated cluster, with an LD of *r^2^ > 0.80*, and, consequently, are associated with BMI at similar significance levels, as well as with other obesity-related traits. 

In the recent analysis, the rs1801282 (in *PPARγ*) was of special relevance. This SNP showed to have a nominally significant effect on reducing the risk for central obesity among Roma, but not in the HG population. On the other hand, this locus was significantly associated with a notable decrease in waist circumference (as a continuous outcome) only in the HR population. This SNP has been described as a risk factor for obesity in Spanish subjects [38], but a non-significant protective effect has been shown in East-Asian subjects [39]. *PPARγ* itself is a significant predictor of the BMI trait, together with lifestyle factors and *FTO* [40]. 

Moreover, *PPARγ* plays a significant role in modulating the expression of genes involved in fat cell differentiation, inflammatory processes, and insulin sensitivity [41,42,43]. The rs1801282 in *PPARγ* has been associated with higher BMI in most of the candidate gene studies, but there are reports that have shown an opposite association, or have not found any significant association at all [38,39,44,45,46,47]. 

The sparse studies used for our meta-analysis (see section ‘Meta-analysis for *PPARγ* (rs1801282)’ in the Appendix A) showed that rs1801282 among HR had a divergent direction of the association compared with that of Indian populations (the alleged origins of Roma). 

Such results may indicate that, if this variant does modulate obesity susceptibility, it may do so in combination with other environmental and lifestyle-related factors. This speculation can be supported by several reports showing interactions between *PPARγ* and factors such as sex, physical activity level, dietary fat intake, total energy intake, and breast-feeding practices on obesity phenotypes [40,48,49,50,51]. In our study, HR and HG populations’ environmental exposures may have differed. However, our current data did not permit exploration of such aspects, and this may be a future research question to be addressed.

In their entirety, our results show that obesity-related SNPs indicate associations of the same direction and magnitude with phenotypes in both populations. It should be highlighted that their effects on waist circumference were consistently stronger compared to those on general obesity for both study groups and, overall, most of the effect sizes and association measures for SNPs examined in this analysis can be sufficiently applied for risk estimation for obesity in HR, except for a locus in *PPARγ*. 

It is intriguing to speculate why effect estimates regarding obesity-relevant genetic loci described for European populations are applicable for risk assessment in the case of the Roma population of hypothesized Asian ancestry. Interestingly, an identity-by-descent (IBD) segment analysis showed that Roma are more closely related to European populations than to Asian ones. The average shared IBD segment length of Roma with Central European populations was 0.355 Mb, while their highest relatedness to Northwest Indian groups throughout India was found with an average shared IBD segment length value of 0.132 Mb, and to the Pakistani groups an average share of 0.087 Mb could be identified [7]. 

Additionally, the most recent data from the second-largest ancient DNA study ever reported on genomic history of South-Eastern Europe [52] suggests that this region has served as a genetic contact zone over thousands of years. This in turn may indicate that major genetic distinctions may have been lost over time due to interaction of population genetics shaped by convergent evolution. Previous work has identified specific immune pathways being shaped by convergent evolution in Roma and European ancestry populations [53]. Taken together, these findings may indirectly suggest that the HR genetic background today might be closer to European genetic architecture than to Asian ones. 

This study had several strengths as it addressed the need to study more diverse populations using both empirical examples and theoretical reasoning. In addition, we provided an accurate comparison with different phenotyping criteria for obesity and a detailed assessment for the replicability of the effect of obesity-related SNPs in the Roma population, which indicated that the effect size measurements obtained from the Hungarian general population can be used for risk estimation for both populations. We also acknowledge that the current research has limitations. A common challenge in ethnicity-based studies is the accurate determination of ethnicity. In the present study, Roma ethnicity status was self-reported and HR subjects were sampled in North-East Hungary, wherein Roma are accumulated in segregated colonies [14]. However, they may not be fully representative for the Hungarian Roma population in general. 

Moreover, more than 8% of the Hungarian population is Roma [9,14] and it is reasonable to suppose that in the HG sample, individuals belonging to the Roma population were also present, which results in a potential slight underestimation of differences between the two study groups. Another important caveat for our study was that the number of subjects in the present study may not have reached the critical sample size, resulting in a reduced study power for SNPs with limited individual effect on outcome. Further work is required to replicate and validate our findings on independent Roma samples for rs1801282 in *PPARγ*, and sexual dimorphism (which determines the patterns of body fat distribution and defines body shape) needs to be further investigated, as this was out of the scope of the present work. 

## 5. Conclusions

In summary, the effects of a relevant obesity-associated gene variants among adult HG and HR populations were reported. The extent to which these associations are attributable to increased general and abdominal obesity were described, accounting for body mass index and waist circumference with different phenotyping criteria. Differences of selected variants on obesity risk among the two populations were fairly modest, except *PPARγ’s* protective effect on central obesity among Roma subjects. Findings suggest sufficient transferability of data related to susceptible SNPs identified in European adult populations to the admixed population of Hungarian Roma. Future studies are needed to assess transferability and applicability of more SNPs related to susceptibility of obesity and other diseases and disorders, especially those that may manifest a more pronounced impact on the health status of these communities. Our findings also highlight the value of surveying across ethnic groups, towards the fine-mapping search efforts for variants, which may be ethnic-specific. Understanding variation in genetic susceptibility across different populations of diverse ancestries might allow better and more targeted prevention for obesity and will enable a better genetic characterization of complex diseases in population groups of interest.

## Figures and Tables

**Figure 1 genes-11-00516-f001:**
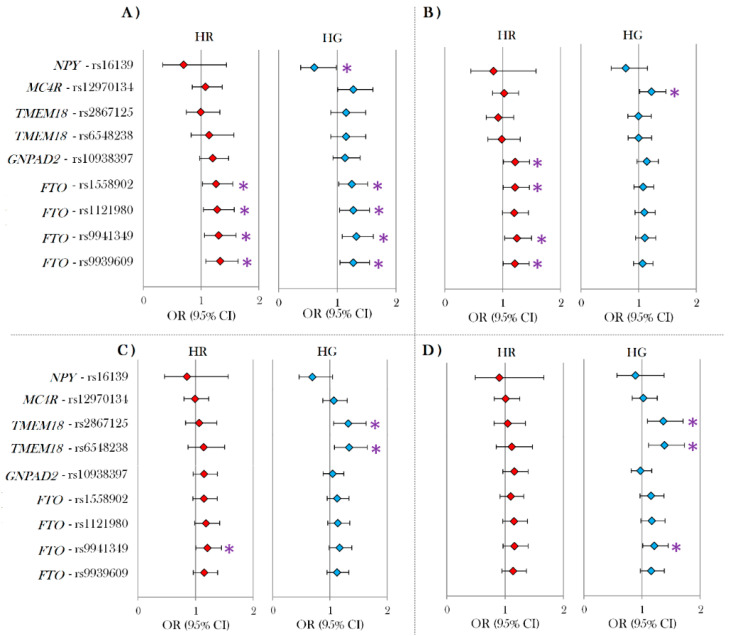
Odds ratios (ORs) of SNPs significantly associated with different obesity-related phenotypes for at least one study population. (**A**) OR for obesity (BMI < 25 vs. BMI ≥ 30); (**B**) OR for WC by ATPIII criteria; (**C**) OR for WC with IDF European criteria; (**D**) OR for WC with IDF Asian criteria. Nominally significant differences are marked with an asterisk (*).

**Table 1 genes-11-00516-t001:** Demographic and anthropometric characteristics of Hungarian general and Roma populations.

Characteristics	Hungarian General (*N* = 1496)	Hungarian Roma (*N* = 1141)	*p*-Value
Females (%) / Males (%)	52.87/47.13	59.10/40.90	0.001
Age (mean value (years) ± SD)	44.17 ± 11.81	40.22 ± 13.00	<0.001
BMI (mean value (kg/m^2^) 95% CI)	28.03 (27.75–28.32)	26.15 (25.69–26.61)	<0.001
WC (mean value (cm) 95% CI)	94.89 (94.14–95.65)	90.09 (89.08–91.10)	<0.001
Abdominal/central obesity (%) ATPIII ^1^	47.79	40.23	<0.001
Abdominal/central obesity (%) IDF_EURO_ ^2^	69.85	56.35	<0.001
Abdominal/central obesity (%) IDF_ASIA_ ^3^	75.47	59.16	<0.001

^1^ Classification based on Adult Treatment Panel III criteria. ^2^ Classification based on International Diabetes Federation criteria for Europeans. ^3^ Classification based on International Diabetes Federation criteria for Asians. SD: Standard deviation, CI: Confidence interval, BMI: Body mass index, WC: Waist circumference.

**Table 2 genes-11-00516-t002:** Effect sizes of single nucleotide polymorphisms (SNPs) showing significant associations with BMI and WC (as a continuous outcome) in the Hungarian general (HG) and Hungarian Roma (HR) populations.

SNP	Gene ^†^	Risk Allele	Phenotype	HG (*N* = 1496)			HR (*N* = 1141)			*p*-Value ^‡^
				*β*	SE	*p*-Value	*β*	SE	*p*-Value	
rs1121980	*FTO*	A	BMI	**0.553**	0.205	0.007	0.604	0.356	0.090	0.888
			WC	**1.516**	0.472	0.001	**1.682**	0.752	0.025	0.679
rs1558902	*FTO*	A	BMI	**0.482**	0.205	0.019	0.628	0.360	0.081	0.697
			WC	**1.318**	0.472	0.005	**1.513**	0.758	0.046	0.730
rs9939609	*FTO*	A	BMI	**0.515**	0.205	0.012	**0.756**	0.357	0.035	0.525
			WC	**1.333**	0.472	0.005	**1.596**	0.754	0.035	0.618
rs9941349	*FTO*	T	BMI	**0.618**	0.206	0.003	**0.770**	0.357	0.031	0.666
			WC	**1.567**	0.474	0.001	**1.546**	0.754	0.040	0.801
rs1801282	*PPARγ*	C	BMI	−0.005	0.314	0.988	−1.188	0.807	0.141	0.177
			WC	0.340	0.724	0.638	**−3.507**	1.699	0.039	0.029 *

Significant effect sizes and associations are highlighted in grey. *Β*: Effect size, linear regression; SE: Standard error of effect size. **^†^** The effect of SNPs in *FTO* is not independent of each other due to the high degree of linkage disequilibrium; ^‡^
*p*-value for difference in association between study population; * nominal significance and ** multiple comparison adjusted significance level for cross-group comparison.

**Table 3 genes-11-00516-t003:** Result of adjusted β value * in genetic models for candidate gene polymorphism *PPARγ* (rs1801282) on the waist circumference as continuous outcome. The most frequent homozygote genotype was considered as the reference (ref.)

*Hungarian Roma*					
Genetic Model	Genotype	Difference (95%CI)	*p*-Value	AIC **	BIC **
Codominant	C/C	ref.	0.140	8928.1	8957.9
	C/G	3.44 (–0.05 –6.93)			
	G/G	4.72 (–4.20 –23.65)			
**Dominant**	**C/C**	**ref.**	**0.048**	**8926.1**	**8950.9**
	**C/G and G/G**	**3.48 (0.04 –6.92)**			
Recessive	C/C and C/G	ref.	0.650	8929.9	8954.7
	G/G	4.38 (–14.57 –23.32)			
Over-dominant	C/C and G/G	ref.	0.054	8926.3	8951.2
	C/G	3.43 (–0.06 –6.91)			
***Hungarian General***					
**Genetic Model**	**Genotype**	**Difference (95%CI)**	***p*-Value**	**AIC ****	**BIC ****
Codominant	C/C	ref.	0.870	11812.4	11844.1
	C/G	–0.25 (–1.86 –1.35)			
	G/G	–1.21 (–6.60 –4.19)			
Dominant	C/C	ref.	0.690	11810.5	11837
	C/G and G/G	–0.32 (–1.88 –1.25)			
Recessive	C/C and C/G	ref.	0.680	11810.4	11836.9
	G/G	–1.15 (–6.53 –4.23)			
Over-dominant	C/C and G/G	ref.	0.780	11810.5	11837
	C/G	–0.23 (–1.83 –1.37)			

* The model was adjusted for age and sex. ** Akaike information criterion (AIC) and Bayesian information criterion (BIC) for three genetic models. The lower the AIC and BIC value, the better the model. Selected genetic model after considering Akaike information criterion for OR (95% CI) and *p*-value < 0.05 is considered significant, OR and corresponding 95% CI adjusted for age and sex as covariates. The *p*-value < 0.05 is represented in bold.

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
