# Peer review of "Applicability of Obesity-Related SNPs and Their Effect Size Measures Defined on Populations with European Ancestry for Genetic Risk Estimation among Roma"

_genes, 2020, doi:10.3390/genes11050516_

Round 1

Reviewer 1 Report

Why analyze all SNPs that were in LD with each other instead of using a representative SNP?  For example, all of the significant SNPs in FTO were in strong LD with each other.  Why analyze all SNPs, which basically represent one association, rather than chose a representative SNP?   As expected they all have similar betas or OR due to the LD, so you are not gaining any information.

What is the significance of the exact same LD blocks in both populations?  Does this strengthen or limit your findings?

The differences in effect alleles do not seem to affect the magnitude of the associations in the 2 populations, with similar effect sizes in both.  Why do you think this is the case?

Author Response

1 - Why analyze all SNPs that were in LD with each other instead of using a representative SNP?  For example, all of the significant SNPs in FTO were in strong LD with each other.  Why analyze all SNPs, which basically represent one association, rather than chose a representative SNP?   As expected they all have similar betas or OR due to the LD, so you are not gaining any information.

The weighted genetic risk (wGRS) calculation requires the knowledge of the effect of SNPs on the outcome. If the effect of SNPs differs in the populations to be studied, the use of the same effect indicators will lead to erroneous results obtained during the wGRS calculation. The different origin of the Hungarian general and Roma populations have raised the question whether the effect values of the SNPs used in our previous study [1] is the same or different in the two study populations.

As the Reviewer pointed out, the distinct pattern of linkage disequilibrium values for four out of the five SNPs in FTO gene are high (R2>0.8). Even though the degree of linkage is very high, one cannot rule out the possibility that there are other ethnicity-specific mutations in the DNA segments bounded by these SNPs (Figure 1 and 2 below) which may affect the effect of these SNPs on the outcome. To eliminate the effect of these unknow factors, we performed the analysis for all SNPs independently from the LD values.

Figure 1: The position of the 5 SNPs in the FTO gene (rs6499640 in purple, rs1558902 in red, rs1121980 in green, rs9939609 in blue, and rs9941349 in black)

Find figure on document attached

Figure 2: Gene variants near the five SNPs in FTO gene

Find figure on document attached

In order to draw the reader’s attention to this phenomenon, an extra sentence has been inserted in the footnotes of Table 4 in the manuscript: “The effect of SNPs in the FTO gene is not independent of each other due to the high degree of linkage disequilibrium”.

2 - What is the significance of the exact same LD blocks in both populations?  Does this strengthen or limit your findings?

The pattern of linkage disequilibrium in a genome is a powerful indicator of the population genetic processes that are structuring it. We used LDmatrix Tool (https://ldlink.nci.nih.gov/?tab=ldhap) to estimate the pattern of LD in both populations. Utah Residents with Northern and Western European ancestry (CEU) were considered as references for the Hungarian general population, while the Gujarati Indian (GIH) population samples were used for the Roma population. In case of the Hungarian Roma population, the pattern of LD was more similar to the CEU than to the GIH (Table 1).

The similarity in LD patterns suggests that the study populations do not differ significantly in the obesity-related set of SNPs in our investigation. These results also support the following statement in the manuscript: “Additionally, the most recent data from the second-largest ancient DNA study ever reported on genomic history of South-eastern Europe [2], suggests that this region has served as a genetic contact zone over thousands of years. This in turn may indicate that major genetic distinctions may have been lost over time due to interaction of population genetics, shaped by convergent evolution. Previous work has identified specific immune pathways, being shaped by convergent evolution in Roma and European ancestry populations [3]. Taken together, these findings may indirectly suggest that HR genetic background today might be closer to European genetic architecture than to Asian ones.

Table 1: The calculated and estimated LD for the Hungarian general and Roma population. The numbers in brackets indicate the values of LD estimated by the LDmatrix Tool online LD calculator. Utah Residents with Northern and Western European ancestry (CEU) were considered as references for the Hungarian general population, while the Gujarati Indian (GIH) population samples were used for the Roma population.

Hungarian general (CEU)

rs6499640

rs1558902

rs1121980

rs9939609

rs9941349

rs6499640

1.00

0.07 (0.11)

0.07 (0.10)

0.07 (0.13)

0.07 (0.12)

rs1558902

0.07 (0.11)

1.00

0.94 (0.92)

0.91 (0.92)

0.87 (0.86)

rs1121980

0.07 (0.10)

0.94 (0.92)

1.00

0.88 (0.85)

0.91 (0.94)

rs9939609

0.07 (0.13)

0.91 (0.92)

0.88 (0.85)

1.00

0.93 (0.90)

rs9941349

0.07 (0.12)

0.87 (0.86)

0.91 (0.94)

0.93 (0.90)

1.00

Hungarian Roma (GIH)

rs6499640

rs1558902

rs1121980

rs9939609

rs9941349

rs6499640

1.00

0.14 (0.01)

0.15 (0.04)

0.18 (0.00)

0.14 (0.02)

rs1558902

0.14 (0.01)

1.00

0.86 (0.71)

0.88 (0.88)

0.84 (0.59)

rs1121980

0.15 (0.04)

0.86 (0.71)

1.00

0.85 (0.62)

0.88 (0.90)

rs9939609

0.18 (0.00)

0.88 (0.88)

0.85 (0.62)

1.00

0.84 (0.70)

rs9941349

0.14 (0.02)

0.84 (0.59)

0.88 (0.90)

0.84 (0.70)

1.00

3 - The differences in effect alleles do not seem to affect the magnitude of the associations in the 2 populations, with similar effect sizes in both.  Why do you think this is the case?

 We understand the Reviewer's concern properly and accept the fact that the effect of the SNPs differs minimally, but not statistically significantly between the two study populations. Based on the results of the present study, we cannot rule out the possibility that the combined impact of these SNPs is significantly affecting the outcome. The reason why we suppose that these not significant but existing difference in the effect of SNPs do not affect the magnitude of the association that in our previous study [1], we examined the combined effect of these SNPs by genetic risk scores (GRSs) calculation and found that there is a significant correlation between the GRSs and obesity-related outcomes after correction to the ethnicity.

References

 Nagy, K., et al., Distinct Penetrance of Obesity-Associated Susceptibility Alleles in the Hungarian General and Roma Populations. Obes Facts, 2017. 10(5): p. 444-457.

  1. Mathieson, I., et al., The genomic history of southeastern Europe. Nature, 2018. 555: p. 197.
  2. Laayouni, H., et al., Convergent evolution in European and Rroma populations reveals pressure exerted by plague on Toll-like receptors. Proceedings of the National Academy of Sciences, 2014. 111(7): p. 2668.

Reviewer 2 Report

Present report from Ádány group enfatize some findings reported in a previous paper (Károly Nagy, Szilvia Fiata, János Sándor, Róza Ádány. Distinct Penetrance of Obesity-Associated Susceptibility Alleles in the Hungarian General and Roma Populations. Obes Facts 10: 444-457; 2017) based on the substantially same population samples.

Present paper enlight that some SNP frequencies of FTO gene, also analysed in the previous work, might be different between Hungarian subjects with a Roma or European ancestry when IDF’s European and Asian criteria are differently applied to define central obesity. In addition a protective effect of PPARγ rs1801282 SNP for Roma subjects was observed.

In their conclusion  authors indicate that differences of selected variants on obesity risk among the two populations were fairly modest, except 408 PPARγ’s protective effect on central obesity among Roma subjects. Moreover Authors confirm their previous paper suggesting that central obesity genetic susceptibility setting of Hungarian European adult populations is not substantially different from the admixed population of  Hungarian Roma.

In this light paper seems do not add any particular piece of evidence to the previous report of the same group and scarcely suitable for publication.

However, I suggest that paper might be substantially shortened and focused on PPARγ rs1801282 SNP, if better analised in light of European and Indian obesity susceptibility genetic background (see suggested references below) This might increase attractive of paper.

Phani NM, Vohra M, Rajesh S, Adhikari P, Nagri SK, D'Souza SC, Satyamoorthy K, Rai PS. Implications of critical PPARγ2, ADIPOQ and FTO gene polymorphisms in type 2 diabetes and obesity-mediated susceptibility to type 2 diabetes in an Indian population. Mol Genet Genomics. 2016 Feb;291(1):193-204.;

Saini S, Walia GK, Sachdeva MP, Gupta V. Genetics of obesity and its measures in India. J Genet. 2018; 97:1047-1071.

Dietrich S, Jacobs S, Zheng JS, Meidtner K, Schwingshackl L, Schulze MB. Gene-lifestyle interaction on risk of type 2 diabetes: A systematic review. Obes Rev. 2019; 20: 1557-1571

Goni L, García-Granero M, Milagro FI, Cuervo M, Martínez JA. Phenotype and genotype predictors of BMI variability among European adults. Nutr Diabetes. 2018: 27. doi: 10.1038/s4 1387-018-0041-1.

Moreover the effect of environmental and dietary factors on Roma genetic background reshaping should be taking in account in discussion on genetic susceptibility to a multifactorial phenotype as obesity (As an example see: Laayouni et al. Convergent evolution in European and Roma populations reveals pressure exerted by plague on Toll-like receptors. PNAS 2014 111:2669).

Author Response

1 - Present report from Ádány group emphasize some findings reported in a previous paper (Károly Nagy, Szilvia Fiata, János Sándor, Róza Ádány. Distinct Penetrance of Obesity-Associated Susceptibility Alleles in the Hungarian General and Roma Populations. Obes Facts 10: 444-457; 2017) based on the substantially same population samples.

Present paper enlightens that some SNP frequencies of FTO gene, also analysed in the previous work, might be different between Hungarian subjects with a Roma or European ancestry when IDF’s European and Asian criteria are differently applied to define central obesity. In addition, a protective effect of PPARγ rs1801282 SNP for Roma subjects was observed.

In their conclusion authors indicate that differences of selected variants on obesity risk among the two populations were fairly modest, except PPARγ’s protective effect on central obesity among Roma subjects. Moreover, Authors confirm their previous paper suggesting that central obesity genetic susceptibility setting of Hungarian European adult populations is not substantially different from the admixed population of Hungarian Roma.

In this light, the paper seems to not add any particular piece of evidence to the previous report of the same group and is scarcely suitable for publication.

However, I suggest that the paper might be substantially shortened and focused on PPARγ rs1801282 SNP, if better analysed in light of European and Indian obesity susceptibility genetic background (see suggested references below) This might increase attractiveness of the paper.

Phani NM, Vohra M, Rajesh S, Adhikari P, Nagri SK, D'Souza SC, Satyamoorthy K, Rai PS. Implications of critical PPARγ2, ADIPOQ and FTO gene polymorphisms in type 2 diabetes and obesity-mediated susceptibility to type 2 diabetes in an Indian population. Mol Genet Genomics. 2016 Feb;291(1):193-204.;

Saini S, Walia GK, Sachdeva MP, Gupta V. Genetics of obesity and its measures in India. J Genet. 2018; 97:1047-1071.

Dietrich S, Jacobs S, Zheng JS, Meidtner K, Schwingshackl L, Schulze MB. Gene-lifestyle interaction on risk of type 2 diabetes: A systematic review. Obes Rev. 2019; 20: 1557-1571

Goni L, García-Granero M, Milagro FI, Cuervo M, Martínez JA. Phenotype and genotype predictors of BMI variability among European adults. Nutr Diabetes. 2018: 27. doi: 10.1038/s4 1387-018-0041-1.

The suggested literature by the Reviewer has been of great use. We have made substantial revisions and re-arranged the Results and Discussion sections (indicated by the track changes) in order to shorten the manuscript and focused on PPARγ rs1801282 SNP as the Reviewer suggests. Some of the information previously described in the main manuscript has been moved to the supplementary material and for the interested reader it has been indicated where this information can be found. Specifically, a new sub-section (3.3 Genetic model for gene polymorphism PPARγ (rs1801282)) has been added to the Results section, with a new table containing information on the PPARG SNP genetic models (Table 3) as described in one of the Reviewer’s suggested references [1] and additional comprehensive tables have been added to the supplementary material regarding the comparison of adjusted odds ratio of genetic models for PPARγ (rs1801282) polymorphism with different phenotyping criteria among HR, as well as among HG group. Further, references recommended by the Reviewer have been considered and included our effort to expand our analyses in the Results section, as well as in enriching the Discussion section.

Additional logistic regression and association analyses were also performed assuming co-dominant, dominant and recessive models. Akaike information criterion (AIC) and Bayesian information criterion (BIC) for each model are presented in the main manuscript for the PPARγ SNP (rs1801282) and more models are made available in the supplementary file (Tables S6-S9). Further, four genetic models (i.e. codominant, dominant, recessive, and overdominant) to further examine the effect of the rs1801282 in PPARγ gene on the WC as a continuous outcome. There was a significant association between SNP and outcome in the Roma population with a dominant genetic model. Furthermore, it was found that the direction of the effect of the rs1801282 differs in the two study populations regardless of the genetic model.

Similarly to one of the works suggested by the Reviewer as a basis, we performed a meta-analysis of the association of the flagged SNP (i.e. rs1801282 [PPARγ]) with obesity phenotype in South Asian and Indian populations – being the region of the alleged origin of HR population (see section ‘Meta‑analysis for PPARγ (rs1801282)’, in the supplementary material).

2 - Moreover the effect of environmental and dietary factors on Roma genetic background reshaping should be taking in account in discussion on genetic susceptibility to a multifactorial phenotype as obesity (As an example see: Laayouni et al. Convergent evolution in European and Roma populations reveals pressure exerted by plague on Toll-like receptors. PNAS 2014 111:2669).

Reviewer’s comment is very relevant, in that we need to account for possible implications of the Gene-Environment (GxE) interactions in our interpretations. The indicated study by the Reviewer is in line with our suggestion in the Discussion section, and it has been used in the expanded version of our statement: “Additionally, the most recent data from the second-largest ancient DNA study ever reported on genomic history of South-eastern Europe [2], suggests that this region has served as a genetic contact zone over thousands of years. This in turn may indicate that major genetic distinctions may have been lost over time due to interaction of population genetics, shaped by convergent evolution. Previous work has identified specific immune pathways, being shaped by convergent evolution in Roma and European ancestry populations [3]. Taken together, these findings may indirectly suggest that HR genetic background today, might be closer to European genetic architecture than to Asian ones.

In addition, we have considered the possible GxE interaction implications and added the following statement in the Discussion after also considering the new findings from the meta-analyses:

‘Such results may indicate that, if this variant does modulate obesity susceptibility, it may do so in combination with other environmental and lifestyle-related factors. This speculation can be supported by several reports showing interactions between PPARγ and factors such as sex, physical activity level, dietary fat intake, total energy intake and breast-feeding practices on obesity phenotypes [4-8]. In our study HR and HG populations’ environmental exposures may differ. However, our current data do not permit exploration of such aspects and this may be a future research question to be addressed.’

Eventually, since data on dietary aspects were not available, it was not possible to account for them in the analyses of the samples under investigation. To emphasize this fact, we have added the following statement (also shown above) in the Discussion section:

‘However, our current data do not permit exploration of such aspects and this may be a future research question to be addressed.’

References

  1. Phani, N.M., et al., Implications of critical PPARγ2, ADIPOQ and FTO gene polymorphisms in type 2 diabetes and obesity-mediated susceptibility to type 2 diabetes in an Indian population. Molecular genetics and genomics, 2016. 291(1): p. 193-204.
  2. Mathieson, I., et al., The genomic history of southeastern Europe. Nature, 2018. 555: p. 197.
  3. Laayouni, H., et al., Convergent evolution in European and Rroma populations reveals pressure exerted by plague on Toll-like receptors. Proceedings of the National Academy of Sciences, 2014. 111(7): p. 2668.
  4. Goni, L., et al., Phenotype and genotype predictors of BMI variability among European adults. Nutrition & Diabetes, 2018. 8(1): p. 27.
  5. Randall, J.C., et al., Sex-stratified genome-wide association studies including 270,000 individuals show sexual dimorphism in genetic loci for anthropometric traits. PLoS Genet, 2013. 9(6): p. e1003500.
  6. Lamri, A., et al., Dietary fat intake and polymorphisms at the PPARG locus modulate BMI and type 2 diabetes risk in the DESIR prospective study. International journal of obesity, 2012. 36(2): p. 218-224.
  7. Verier, C., et al., Breast-Feeding Modulates the Influence of the Peroxisome Proliferator–Activated Receptor-γ (PPARG2) Pro12Ala Polymorphism on Adiposity in Adolescents: The Healthy Lifestyle in Europe by Nutrition in Adolescence (HELENA) cross-sectional study. Diabetes care, 2010. 33(1): p. 190-196.
  8. Dedoussis, G.V., et al., An age-dependent diet-modified effect of the PPARγ Pro12Ala polymorphism in children. Metabolism, 2011. 60(4): p. 467-473.

Round 2

Reviewer 2 Report

The substantial revisions and re-arrangement of the "Results and Discussion" sections meet my previous major suggestions. Focusing on PPARγ rs1801282 SNP data Authors have improved manuscript scientific significance and interest of the paper.